# Everyday Experiences of People Living with Mild Cognitive Impairment or Dementia: A Scoping Review

**DOI:** 10.3390/ijerph191710828

**Published:** 2022-08-30

**Authors:** Jacoba Huizenga, Aukelien Scheffelaar, Agnetha Fruijtier, Jean Pierre Wilken, Nienke Bleijenberg, Tine Van Regenmortel

**Affiliations:** 1Institute of Social Work, HU University of Applied Sciences Utrecht, 3507 LC Utrecht, The Netherlands; 2Research Center Social Innovation, HU University of Applied Sciences Utrecht, 3507 LC Utrecht, The Netherlands; 3Department of Tranzo, School of Social and Behavioral Sciences, Tilburg University, 5000 LE Tilburg, The Netherlands; 4Research Center Healthy & Sustainable Living, HU University of Applied Sciences Utrecht, 3507 LC Utrecht, The Netherlands; 5Department Julius Center for Health Sciences and Primary Care, University Medical Center Utrecht, 3584 CG Utrecht, The Netherlands; 6HIVA—Research Institute for Work and Society, Faculty of Social Sciences, University of Leuven, B-3000 Leuven, Belgium

**Keywords:** dementia, mild cognitive impairment, lived experience, everyday life, citizenship, living in the community

## Abstract

Increasing attention has been paid to the ‘voice’ of people living with mild cognitive impairment (MCI) or dementia, but there is a lack of clarity about how everyday life is perceived from this insider’s perspective. This study aimed to explore the everyday life experiences, challenges and facilitators of individuals with MCI and dementia living at home. A scoping review of qualitative studies, guided by the Joanna Briggs Institute Reviewers Manual, was conducted. Eight databases were searched, resulting in 6345 records, of which 58 papers published between 2011 and 2021 were included. Analysis was carried out by descriptive content analysis. Findings were categorized into seven spheres of everyday life: experiences related to the condition, self, relationships, activities, environment, health and social care and public opinions. The results show many disruptions and losses in everyday life and how people try to accommodate these changes. In all areas of everyday life, people show a deep desire to have reciprocal relationships, stay engaged through participation in activities and have a sense of belonging in the community. However, more research is needed on the factors that promote and impede the sense of reciprocity and belonging.

## 1. Introduction

The umbrella term ‘dementia’ describes dementia as a chronic and progressive condition of deterioration in cognitive function which affects daily functioning (DSM-5). Currently, there are approximately 50 million people worldwide with dementia, and recent data estimate that the prevalence of dementia will triple by 2050 [1]. Because people with dementia live longer at home within their social networks, the rising number of people with dementia will become one of the greatest challenges for health, social care and society [2].

In Western societies, the discourse on dementia care and research has long dominated by a biomedical view, focusing mainly on what is lost, also referred to as a ‘deficit’ model [3]. While this approach has benefits, such as an emphasis on diagnosis, pathogenesis and pharmacological treatment, it is not without problems. The first is the tendency to focus only on an illness that needs to be ‘fixed’, with little attention to the psychosocial aspects of living with dementia [4]. The second problem is that it positions people living with dementia as ‘patients’ and therefore limited agency is given to them compared to their caregivers. In addition, the biomedical focus has also been associated with stigmatizing views society and media have developed regarding ‘demented’ people, i.e., people that are affected by and are victims of this disease [5]. Consequently, assumptions about the lack of remaining abilities and qualities arise, risking persons with dementia being treated as the ‘living dead’, and thereby actually giving up on them [6]. It can be argued that Western society historically tends to move towards a deficit model, a society that Post called the ‘hypercognitive’ culture, in which dementia is to be feared because a lack of cognitive capacities violates self-control and independence [7]. The ‘discourse of loss’ is the loss of ‘self’ [8] which can be traced back to the legacy of the Enlightenment, i.e., the duality of body and mind [9,10].

To extend beyond the understanding that dementia is only a biomedical problem, Kitwood [11] developed his theory of ‘malignant social psychology’. This refers to the influence of behaviours of the environment that undermine the personhood of people living with dementia. Consequently, people are misunderstood, marginalized, stigmatized and even mistreated [12,13]. Kitwood emphasized the role of meaningful relationships in the well-being of people living with dementia [11]. This emphasis formed the basis for the development of person-centred care, which aims to strengthen the personhood of people through a supportive social environment [14,15]. This emerging perspective on care has led to a fundamental paradigm shift in dementia research and practice [16]. Dementia is increasingly viewed as a mutual interaction between the biological, psychological and social domains [17]. In contrast to the body-mind duality, Kontos introduced the notion of the lived body and embodied selfhood of persons with dementia [18,19,20], influenced by the work of Merleau-Ponty. There has been a growing interest in person-centred interventions, for example creative and arts-based interventions that focus on the remaining strengths of people and redress the focus on the deficit [21,22,23,24]. These interventions improve outcomes such as subjective well-being, autonomy and the quality of relationships [25,26,27]. A recent systematic review and meta-analysis shows that person-centred interventions positively reduce behavioural and psychological symptoms and improve cognitive functioning [28].

Another paradigm that affects the discourse in the field of dementia is the concept of social citizenship. This concept fits within the development of approaches in other fields such as the recovery movement in mental health care and the human rights movement by people with disabilities. Recovery refers to taking back control of one’s life and illness [29,30]. The empirically grounded CHIME framework compromises five key components of recovery: Connectedness, Hope, Identity, Meaning and Empowerment [31]. In the field of disabilities the social model of disability focuses on the exclusion of people and the need for human rights [32]. This is clearly reflected in the United Nations Convention on the Rights of Persons with Disabilities (UNCRPD), adopted on 13 December 2006: “persons with disabilities should have the opportunity to be actively involved in decision-making processes about policies and programmes, including those directly concerning them” [33]. Increasing attention is being paid to studying citizenship for people with disabilities [34,35,36]. In research on dementia Bartlett and O’Connor [37] suggested a shift from personhood to citizenship in which people with dementia are positioned as active agents with rights rather than only persons in need. The social citizenship approach stresses the capacity for agency and the right to influence and control decisions that affect their lives and wellbeing [37]. This broader lens recognises that people are embedded in and shaped by a sociocultural context with dimensions of social locations and risks that precede dementia, such as socio-economic inequalities. Consequently, a shift from the person with dementia to society at large is needed. The theory of social citizenship points out that people with dementia are equal citizens who desire to continue participating in society [38,39,40]. The socio-political discourse of deficit has been criticized stigmatizing people living with dementia and not recognizing them as citizens with rights [3]. Dementia should be regarded as a human rights issue [41,42]. In the United Kingdom, the rights-based approach has been embedded in the Dementia Statements [43]. A qualitative meta-study identified how contextual forces such as access to places enabled or hindered social citizenship in the everyday lives of people with dementia [44]. Therefore, to understand how to support people with dementia towards living well as citizens, more emphasis on the mundane aspects and settings of people’s lives is needed [45]. Moreover, this requires working on an inclusive society [9], so that people with dementia can continue to engage in the world outside their homes [46,47].

The way of viewing dementia and people living with dementia influences how people are approached as well as the way interventions are shaped. Much research has been conducted on health and social care interventions for people living with dementia and their families [48,49]. Nevertheless, services are suboptimal in meeting the needs of people with dementia and their families [50,51]. Research supports this notion and reveals that the effectiveness of psychosocial interventions is limited [52]. Critics of research on interventions state that the chosen outcomes of interventions are highly variable and not based on what individuals living with dementia value in their lives [53,54]. In addition, in intervention research the concept of everyday life is often ‘bracketed’ because of a focus on different aspects such as behaviour problems or care burdens [52]. As a response, and in line with the person-centred and citizenship approach, research to gain an insider’s perspective of day-to-day experiences is needed to optimize tailored interventions. Although increasing attention has been paid to the ‘voice’ of people living with dementia, much is still unknown about the perspective of people with dementia on living with dementia in everyday life. Everyday life can be regarded as the mundane and routine aspects of human life [55]. A comprehensive conceptualization of how everyday life is lived and understood from an insider’s perspective is lacking. A focus on the mundane is important in the context of people living with dementia because everyday life can become a major challenge for them [56]. To date, there has been no review that considered everyday life from the perspective of people living with dementia at home.

This scoping review aims to explore the everyday life of people with dementia living at home from an insider’s perspective. To this end we explored the extent, range and nature of the existing literature to gain insight into the experiences of individuals living with dementia in everyday contexts, how they experience themselves, their relations, their position in the community and what challenges and supports them.

## 2. Materials and Methods

A scoping review of qualitative studies was conducted guided by the Joanna Briggs Institute (JBI) Reviewers Manual [57]. The JBI approach to conducting and reporting scoping reviews is congruent with the PRISMA-ScR checklist [58]. A scoping review is useful for an exploratory approach to map and synthesize current knowledge on a broadly defined topic such as everyday life [59]. The research question that guided this scoping review was: what are the everyday experiences of people living with dementia in the home context?

### 2.1. Inclusion and Exclusion Criteria

Inclusion and exclusion criteria were determined in accordance with the ‘PCC’ mnemonic—Participants, Concept, Context, which is recommended for scoping reviews [57]. We included individuals at any stage of dementia and pre-dementia, such as mild cognitive impairment (MCI). MCI is believed to be a pre-dementia stage and is defined as an objectively determined cognitive impairment, that does not meet the criteria of dementia [60]. The review considered everyday life and its synonyms, such as daily life or day-to-day experiences. In addition, related concepts that provided insight into daily life, such as ‘life world’, ‘living with’ and ‘experience’ were included. Daily life encompasses different life domains, such as work, leisure and relationships. Therefore, any article stating that the investigated domain was related to the day-to-day experiences was included. Study types that met the inclusion criteria were empirical qualitative studies in peer-reviewed journals to capture participants’ subjective experiences in everyday life. Qualitative studies are likely to be useful for exploratory questions describing experiences [61]. Qualitative parts of mixed-method studies were also included. A date range filter from 2011 to 2021 was selected because of the volume of literature published in this area and to obtain a more contemporary insight into day-to-day experiences. The finalized inclusion criteria are listed in Table 1.

### 2.2. Search Strategy

A search strategy was developed with an experienced information specialist (AF). A three-step search strategy was utilized in this review. First, an initial investigation resulted in ten ‘pearl papers’: highly relevant papers that ideally should be included in the results of the search string. This selection of ten pearl-papers papers was used to test the viability, sensitivity and specificity of the search string. Four sets of search terms were used based on Participant/Concept/Context/Study design, thereby aligning the search terms with the inclusion criteria. The search terms are listed in Table 2.

For PubMed, the search-strategy was adapted to include a search within words contained in the titles-, abstracts- and index terms of relevant articles. Thereafter, the search strategy was adapted to work for each separate database included in the current review. The following eight databases were searched: PubMed, CINAHL, Web of Science, Embase, PsychInfo, Social Services Abstract, Sociological Abstracts and Social Care Online. The database search was undertaken on 25 August 2021. Finally, a supplementary manual search was performed, based on the reference lists of the included articles to ascertain the thoroughness of the search.

### 2.3. Study Selection

Following the search all identified citations were extracted from the databases and uploaded into EndNote (version X9.3.3). Duplicates were removed using the automated deduplication function in SR-Accelerator (version 2.0) [62]. Eleven papers were screened and discussed by the whole author team to ensure sufficient inter-rater reliability would be achieved. Pilot tests screening titles and abstracts against the inclusion and exclusion criteria were conducted by two independent reviewers (JH and LvB) using Rayyan. Kappa was calculated at 0.77. Given the Kappa statistic of 0.77, (i.e., above recognised acceptable levels of 0.7) double sifting was not deemed necessary.

To increase the efficiency and quality of review screening, the first reviewer (JH) subsequently screened titles and abstracts using ASReview (version 0.18), an active learning software program [63]. At some point in the active learning process mostly irrelevant research remains and therefore a stopping criterion was used when the last 50 reviewed papers were considered irrelevant. As a safeguard a random subsample of 20 per cent was screened by the second reviewer in Rayyan (LvB). The results were compared in Excel and Kappa was calculated at 0.8 (a finding of 0.8 or higher is commonly qualified as a strong level of agreement). Papers yielded by the second screener, which were not included by the first screener or ASReview, were included if deemed appropriate after discussion with the first screener.

Potentially relevant sources were retrieved in full, and their citation details were imported into Excel. Full texts of the selected citations were assessed in detail following the inclusion and exclusion criteria by three reviewers (JH, LvB and AS). Every paper was reviewed by at least two reviewers and all were reviewed by the first author (JH). Disagreements on study inclusion were resolved through discussion and the other authors were periodically involved in these discussions. Reasons for the exclusion of full-text sources of evidence were recorded.

### 2.4. Data Extraction and Analysis

Data were extracted from each included study by one reviewer (JH), using Microsoft Excel. The completed form was checked for accuracy (LvB). Any implications for the review, such as the planned approach to extraction and analysis, were discussed by all the co-authors. The included articles were charted into a table using the following information: author and year of publication, country, focus of the study, participants and method of data collection. The experiences of people who participated in or were studied in the included articles were grouped into categories using the ecosystem health map developed by Barton, incorporating relational and community perspectives [64]. The first version of the identified categories was discussed by the full team and refined using the combined clinical and academic knowledge of all authors. Subsequent category versions were discussed by the first and second authors. After creating categories, the first author coded all findings using Atlas.ti 9.1.5.0 (software for analysing qualitative data). In cases where differences were found between studies that specifically focused on groups of people, for instance people with MCI and dementia, people with young-onset dementia and older adults, people of Western and non-Western countries and people living alone or with a partner, this was separately mentioned in the findings.

## 3. Results

### 3.1. Selection of Evidence

Based on the database searches, 6345 records were identified, and 10 records were identified through citation searching. After removing duplicates, 2391 records were screened by title and/or abstract for relevance. Finally, 58 studies were included in the analysis. An overview of the data selection process is shown in the PRISMA flow diagram [65] in Figure 1.

### 3.2. Characteristics of the Evidence

The characteristics of the included studies are summarized in Table 3 (for all characteristics of included studies, see Appendix A).

The 58 articles included in this review are based on 54 primary studies. As shown in Figure 2, there appears to be an increase in the number of publications in the last three years. Forty-four of the included studies were conducted in Europe with the majority conducted in Scandinavia (n = 21) and the United Kingdom (n = 18). An overview of the subtype of dementia and stage within dementia is not described as this was not always specified in the included studies. Most samples were more likely to have included people with Alzheimer’s Disease and mild to moderate dementia. However, when data on subtype and stage were available, they are provided in Appendix A. One study specifically focussed on the experiences of people living with behavioural-variant frontotemporal dementia. Social determinants, such as ethnicity, urban or rural, socio-economic status and educational background of participants, have rarely been reported. It should be noted that none of the 58 included studies used a definition of ‘everyday life’. The extent to which studies described everyday life varied greatly both in terms of breadth and depth. Interviews and focus groups were sometimes supplemented with more diverse and creative methods, such as walking interviews (n = 6), observations (n = 5), photovoice (n = 3), art-based data (n = 2), home tours, social network mapping, action research groups, diaries and Twitter (n = 1 for each method).

### 3.3. Descriptive Qualitative Content Analysis

Based on the JBI scoping review guidelines [59] to illustrate and summarize the main findings, the results are classified according to the research question and comprised into seven spheres. The seven spheres are listed in Table 4.

The results are presented as a narrative summary supported by a systematic overview (Table 5).

#### 3.3.1. Sphere 1: Dementia Condition

Most studies (83%, n = 43) describe experiences regarding the dementia condition. We identified three subcategories: 1. the experience of receiving the diagnosis as a major life event, 2. changes related to the diagnosis of dementia and the way participants dealt with these changes and 3. how participants did experience their future life with dementia.



Receiving the diagnosis

In twenty-two studies, experiences of receiving a diagnosis of dementia or MCI are described. Participants in these studies responded differently to receiving the diagnosis. The first response concerned shock, confusion, sadness and fear [71,76,80,86,99,101,122,123]. Respondents found it difficult to accommodate the news. Thoft and Ward [116] report that participants felt alone as it was difficult to accept that there was no cure for the disease. Two studies mention that participants were afraid of becoming ‘second-class citizens’ who others would look down on [84,95]. Another three studies mention participants who did not accept the diagnosis or denied the outcome [68,80,123]. Participants in two studies mentioned that, given the frequently experienced shock reaction, it could be important for professionals to not provide all information about the disease immediately after giving the diagnosis [84,101]. In one study participants with young-onset dementia reported that they were initially wrongly diagnosed several times [84].

After receiving the dementia diagnosis, participants in six studies indicated that one dilemma they faced was whether to disclose their diagnosis to others. Disclosure was found to be difficult as they were afraid that people’s attitudes toward them would change [71,76,102,123]. The advantages of being open were receiving positive responses, strengthening current relationships, being able to tell people about the difficulties they faced, and raising broader awareness of dementia [71,76,99,116].



2.Changes and dealing with changes

In 26 studies the topic of change and the way participants dealt with changes are described. In 11 of these studies transition periods emerged, most often starting with a period of grief followed by gradual adjustment and acceptance over time, sometimes referred to as a ‘journey’ [71,74,78,84,86,97,99,105,106,120,123]. The diagnosis was described as useful in three studies, as it explained their behaviour and the feeling that something was wrong [100,103,122].

Several everyday life changes related to dementia have been reported. One broad category of change refers to cognitive decline, such as concentration difficulties and forgetting daily events, names of people and places, people’s faces, birthdays and retrieving stored items [68,70,71,73,86,88,92,95,100,102,103,114,116,117,119,122,123]. Two studies report that participants were aware that their ability to learn had decreased; and therefore, newer tasks were harder to remember [95,116]. A substantial number of studies report on challenges in communication: participants did not find the right words, repeated themselves in conversations, forgot what they wanted to say, or found it difficult to follow conservations as their focus was reduced [71,77,88,95,100,101,111,116,118,121,122]. This was illustrated by one of the participants as follows: “Sometimes I notice that I don’t give the right answers, you know, stray from the theme. That makes me terribly insecure. I can’t trust myself” [122] (p. 9).

A wide range of ways of dealing with cognitive changes are also reported. Compensatory strategies, such as taking more time, repeating, trying to think systematically and making notes are mentioned [73,88,99,100,116,120,122]. However, the strategies were not always effective; for example, it could be difficult to find the notes again. Several tools have been used, such as calendars, whiteboards and technical aids, sometimes with the support of relatives [81,88,98,100,102,105,108,116,120,122,123]. In four studies participants made a specific effort to keep the brain active by reading and memorizing poems [79,85,92,116]. However, when there was no improvement, they experienced frustration and felt that they were losing their control [89]. Coping strategies are mentioned in all 26 studies, including trying to ignore or hide difficulties, using humour, using creative expressions, avoiding stress, maintaining daily structure, staying active, focusing on what they can still do, finding meaningful ways to (re)engage and seeking social and religious support. In several of these studies, a shift in attitude to what was important for them in life is described, for example by paying more attention to ‘little things’ and living in the ‘now’ [67,73,83,86,88,89,95,101,102,105,110,116,122]. One of the participants voiced this concern as followed: “Take it as it is and live today” [100] (p. 276). Four studies report a more negative outlook in that everything would get worse [71,89,98,107]. However, in one of these studies, participants wanted to choose not to give up fighting dementia, and as a result, they joined efforts to fight for equality for people with dementia [71].



3.Future living with dementia

In 17 studies participants shared how they faced their future with dementia. Several studies mention a deep sense of uncertainty about the future, associated with anticipated loss and worries about the trajectory of the disease [68,69,70,71,74,87,88,95,99,103,105,109]. Some of the participants were able to face this and remained positive, while others found this very difficult and constantly felt a shadow of fear. Participants with MCI could feel distressed when experiencing memory problems in everyday life if that could be related to progression to dementia [67,98]. One study reports that for participants with young-onset dementia the uncertainty could be more difficult to endure than dementia itself [105]. Seven studies describe that participants proactively made plans and took control by, for example, moving to a new dwelling or deciding whom to leave their money or inheritance [79,86,96,102,110,122,123].

#### 3.3.2. Sphere 2: Sense of Self

Almost three quarters of the studies (72%; n = 42) include content regarding the sense of self. These experiences can be divided into three subcategories: 1. self-evaluations, 2. emotions and 3. sense of body.



Self-evaluations

In 32 studies participants evaluated themselves as persons. In two studies participants expressed that after diagnosis they felt like different people, or they were concerned that they would turn into a different person [71,92,103]. Several studies describe how participants could re-define their identity, although this was a tough process [68,71,76,105,106,117]. Clemerson report that participants sought knowledge in the reappraisal process, for instance by reading about the disease [76]. Nonetheless, several studies stress that in essence they were still the same [68,71,94,105,107], for example, in one of the studies a participant expressed: “I’m still me, or at least a version of me” [71] (p. 6).

A majority of studies report challenges related to the way participants experienced themselves and their positions. Losing abilities and memories were a threat to their identity which could lead to a sense of self-doubt, uncertainty, and uselessness [66,71,73,76,79,84,89,92,98,122]. As a result of losing competencies, participants also experienced a loss of social roles [76,79,80,83,104,115]. Two studies mention that for some participants these losses were related to other losses they experienced during their lives [79,106]. The experiences of losing autonomy, freedom and control in daily life have been mentioned in a few studies [66,117,120,122].

What was important for participants and what supported them in their identity are described in 31% of the studies (n = 18). The importance of remaining independent and making one’s own decisions for as long as possible has been frequently reported [78,94,95,103,106,109]. A comment from a participant demonstrates this: “Even if I have Alzheimer’s I want to do as I want, go to stores and do what I want” [95] (p. 913).

Accepting support from others could support this sense of autonomy [123]. Participants had a deep desire to be directly involved in dealing with limitations in their daily lives, which provided a sense of control and agency [74,76,111,122,123]. Three other supportive strategies are identified. Firstly, through emphasizing their worth by mentioning positive traits or their significance to others [72,109,114]. Secondly, some studies report that biographical places, events and personal biographies were supportive of identity, for example, a neighbourhood where participants had lived for a long time [75,96,104,114]. Thirdly, being able to make a meaningful contribution, remaining useful and being valued emerged as important [109,120]. Talbot report that participants regained this by becoming experts by experience [115].



2.Emotions

Eighteen of the included studies report experiences of emotions. In six studies participants felt that they changed emotionally, as they sometimes lost control of their emotional reactions such as crying, anger or laughter [71,81,89,105,117,118]. This could also lead to feelings of shame. In response to disruptions in daily life and failures, such as home accidents, negative emotions such as frustration, anger, depression and for some even despair are described [76,77,89,98,100,111,119]. Another frequently shared emotion, was increasing fear and uncertainty when going outside, or the fear of losing control related to further cognitive decline [72,75,79,100,101,103,117,123]. In one of the studies this is highlighted by this quote: “I am afraid to meet the new day. You know, nobody knows what everyday life will bring, but I am not able to handle it like I did before. I am very scared and much misunderstood” [117] (p. 885).



3.Sense of body

Body experiences are reported in 21 studies. This includes becoming more aware of their body, on which they could no longer rely, and consequently, they were no longer at ease with their body [85,122]. It is also described as a felt conflict between the chronological age and the aging body [76,79]. Several studies describe participants feeling tired or exhausted, as coping with their situation costs a lot of energy [72,86,95,100,117,122]. Five studies report a loss of initiative, lack of desire and apathy [72,83,86,88,93], and three studies mention trouble with sleeping or maintaining a day-night sleeping routine [95,119,122]. The rhythm of daily life seemed to slow down, and body movements and pace became slower [114]. In a number of studies, participants experienced bodily complaints or comorbid conditions which in turn affected their everyday lives [73,74,78,80,83,89,95,96,122].

In addition, being healthy, following a diet and engaging in physical activity are also mentioned [72,73,74,93,96,102].

#### 3.3.3. Sphere 3: Relationships

Most of the included studies (n = 46) describe experiences with relationships. These experiences are divided into relationships with a partner and family, friends and community. Relationships are generally described as important sources of connection and support. The absence of relationships, especially for participants living alone, led to feelings of loneliness, which worsened after the diagnosis of dementia [78,97,100,109,111,114,117]. This sense of loneliness could even be existential, as quoted by one participant: “So I feel … alone in the whole world sometimes. Although I know that I’m not, that’s what I feel like” [114] (p. 152). One study report that reminiscing helped overcome feelings of loneliness [78]. Furthermore, four studies mention participants who shared feelings of social exclusion due to their memory problems or age [85,98,117,122].



Relationship with partner and family

In 35 studies experiences in relationships with close partners are described. Several studies specifically describe the importance of social support from partners [69,72,75,78,79,80,93,97,99,106]. Support could be practical, such as driving or helping to remember things, and emotional during times of frustration. In four studies participants stated that they felt dependent on their partner or anxious when their partner was not around [105,116,118,122]. The feeling of thinking at different levels could lead to challenges in communication and slowly drifting away [67,103,122]. Changes in sexual life are only mentioned once [119].

Family is described as an important source of feeling connected and supported. Three studies specifically mention daughters as being supportive [73,101,123]. Family members can also be advocates to overcome challenges in society [94,123]. One study report that participants shared how relationships with close family members improved in an emotional way [122]. The importance of reciprocity in family relationships is highlighted in six studies, for example by helping family members or spending time with their grandchildren [68,77,111,116,120,122].

Nevertheless, challenges in familial relationships have also been reported, most frequently the experience of family members being too protective and disempowering by questioning their capabilities [67,69,93,94,95,98,106,111]. In addition, participants reported feeling that they had lost their meaningful role within their family [83,88,104,116,122]. At the same time, participants emphasised their concerns about being a burden to their family [73,89,92,100,105,123]. Other experiences described are difficulties relating to physical distance [75], avoidance by family members from talking about dementia [84,88] and struggles of their children to accept or believe the disease [89,99,117].



2.Relationships with friends

In 33 studies participants reported about relationships with friends. It is important to feel connected with close friends without feeling stigmatised [67,79,102,122]. Some participants had good contact with existing networks [75,95,113]. Others tried to find new contacts [97], which could be complicated after moving into a new neighbourhood [96,100]. Internet technology could bridge long distances [75]. In two studies participants preferred that people visit them at home instead of visiting others [112,114]. In several studies, participants appreciated gaining new friends through dementia-specific groups, both in-person and online [70,71,76,79,99,115,116,121]. They experienced these relationships as more equal, supporting and empowering, as it implied an ‘unspoken understanding’, which felt like a form of relief [85,101,120,122]. In two studies with participants living alone, they reported that they needed to put an effort into maintaining social contacts, some with the help of their children [79,97].

Nine studies report that participants had lost their friends [76,80,92,97,104,106,117,120,123]. In several studies participants shared that they did not feel understood in their friendships when sharing their difficulties because of relativizing comments or avoidance of the issue [66,88,97,99,117,122]. In three studies with participants with MCI they chose to withdraw themselves as a way to avoid embarrassment due to their mistakes or because of negative peer responses [89,98,103].



3.Relationships in the wider community

Fourteen studies describe relationships in the wider community. Six studies describe how neighbours provided practical support, such as the management of household waste, looking after pets and gardens, or simply having brief social encounters [75,79,97,99,108,113]. In three studies participants informed their neighbours about the illness which provided an understanding and a sense of security [113,117,121]. Six studies describe the neighbourhood as a site for spontaneous encounters, especially if participants lived there for a long time [75,88,96,97,113,123]. For example, they referred to encounters with staff in shops or being greeted by passers-by. Participants actively sought possibilities for connection, such as sitting on a bench in the centre of the neighbourhood or walking their dog. One study also mentions acts of kindness by people in the neighbourhood, such as when they got lost [66]. Furthermore, religious communities provided a sense of community [102]. In two studies challenges in the community are reported, namely the experience that people had a ‘fear’ surrounding dementia [83] and that they felt anxious about making mistakes in public, for example when withdrawing cash or afraid of becoming a victim of crime [66].

#### 3.3.4. Sphere 4: Activities

Most of the included studies (n = 50) describe information regarding activities. Participants shared the importance of staying engaged in daily life through participation in activities, which provided a sense of meaningfulness, independence and belonging. However, the progressive reduction in functioning could lead to inactivity and loss of meaning in activities. Activities can be divided into activities of daily living and participation.



Activities of daily living

In 38 studies, experience with activities of daily living is reported. Two studies describe participants still being capable of selfcare activities [94,111], while another two reported challenges, such as forgetting medication, or forgetting to eat and drink [108,114]. The assistance of relatives, or tools such as schedules and notes is reported to be supportive [74,82,111,118]. In seven studies participants expressed that they experienced taking care of household chores, such as cooking, cleaning and taking care of pets, as meaningful [74,77,78,82,87,101,109]. By doing these activities they maintained their routines, and sensed reciprocity and responsibility. Sixteen studies describe that the household became increasingly difficult to manage and tasks took longer to complete due to forgetting tasks and where things were put, as well as difficulty in recognizing how to use everyday objects [68,78,81,82,87,88,93,99,100,101,102,108,114,118,119,123]. Challenges with cooking in particular are mentioned and some participants consequently stopped for safety reasons. Some, however, did find practical solutions such as timers, preparing uncomplicated food or using meals-on-wheel services. Regarding administrative tasks, such as paying bills and organizing mail, in one study participants articulated their capabilities [94] and in another study, participants adapted the tasks [81], but in four studies participants felt that they lost control over it [82,88,108,119]. For example, they were worried about doing something wrong, so they checked their work over and over.

The activities of daily living outside, such as getting out of the house or going to a shop, are described in eight studies as important for participants [69,75,78,79,87,96,112,121]. This provided a sense of independence and being connected to, and part of society, which in turn prevented feelings of loneliness. In 14 studies, challenges with routine activities outside, such as the effort it takes to prepare for going outside, activities that are experienced as ‘too busy’, dealing with money outside, keeping an overview of the actions that make up an activity, and not getting lost are mentioned [69,75,81,86,88,93,94,96,101,112,113,116,117,122]. In one study participants mentioned the JAM (‘just a minute’) card as useful, to show when they need ‘just a minute’ [94].

In several studies participants adapted their driving routines or had to give up driving for safety reasons [79,80,82,83,84,88,90,102,106,109,123]. In one study participants articulated their capability to drive [94]. Participants who had lost their driver’s license experienced this as a loss of their valued freedom, which also impacted their participation in activities. Experiences with public transport were different: in a few studies participants were happy to use it nearby [75,112], but in ten studies participants felt overwhelmed by the transport system or were afraid to get lost, and therefore only travel accompanied [75,78,79,83,84,93,95,100,108,122].



2.Participation

In 18 studies experiences with work and volunteering are reported. In seven studies participants who were still employed, sometimes in an adapted job, experienced increasing difficulties with more cognitively demanding aspects of their work [84,86,98,101,102,104,119]. Participants in three studies felt employers forced them to give up their work [80,94,122]. In ten studies participants who had to quit working felt a deep sense of loss and strongly missed their daily structure, engagement, role in life and contact with colleagues [80,84,86,88,92,99,101,110,119,120]. In one study a participant expressed: “If we go to our relative’s house, I see how others are active. This breaks my heart… [crying]…I don’t act like an active person: I who could move mountains! When I was employed, I worked from 8 am until 12 midnight; now, why should I be like this? Why?” [92] (p. 3036).

Volunteering provided participants with a new way of contributing to and connecting with the community [70,83,101,102,115,120,122]. A specific way of meaningful volunteering was dementia advocacy, such as speaking at dementia conferences, using Twitter and joining research projects.

Thirty-two studies describe experiences with leisure activities. Several of these studies describe this as social, cognitive and physical engagement, for experiencing meaningful days and reducing stress [67,70,77,89,95,102,104,116,119,120]. Participants wanted to focus on activities they had always done for as long as possible, and discover new activities [69,70,78,104,116,119,122]. A wide range of leisure activities are mentioned: watching TV [88,89,90,97,101], reading books [78,88,102,121], gardening [101,102], creative and cultural activities such as painting and singing in a choir [67,70,77,90,91,99,102,122] and physical activities such as walking and doing exercises [66,67,68,72,74,83,93,95,96,101,102,104,113,116,120,122]. In five studies participants stressed that they liked to learn new things, either individually or with other people with dementia, such as learning to write, knit or use a tablet to play online games [81,101,102,116,122].

At the same time, in 15 studies people reported that they had to reconcile themselves with downsizing or giving up hobbies and activities, for example because of difficulties in concentration, memory problems or physical complaints [66,68,72,80,81,83,88,91,93,98,102,114,116,122,123]. Other specific hindrances described are the feeling of being excluded [123], the absence of group activities [83,96], or difficulty to pay fees for activities [95]. Loss of activities can lead to isolation [83,116].

#### 3.3.5. Sphere 5: Environment

Information regarding the (physical) environment is described in 21 of the 58 studies. Generally related to place, a few studies reported the need for a sense of familiarity with spaces and routines which enabled belonging [69,75,91]. Some people related unfamiliar experiences to existential feelings of disorientation of ‘not-being-at-home’ [122].



At home

Ten studies describe experiences in the home environment. Six studies describe that participants most often stayed at home, and that this sense of security, where things were familiar, became increasingly important [81,82,90,106,111,114]. Participants wanted the home to be safe to navigate, also at night, as well as cosy and surrounded with cherished belongings. Days at home can be monotonous and isolated [83,114]. Looking out through their window or from their balcony to see children or other people passing by gave a sense of connection to the world outside [78]. Four studies describe the transition of moving to a new, smaller house as difficult for various reasons, such as packing in feeling like a too daunting task, or not remembering where things were in the new house [90,96,106,117].



2.Public space

Fifteen studies reported experiences with public spaces. Six studies describe that to feel safe without getting lost, participants preferred familiar spaces close to their home and the routine of going to the same recognizable places by using the same path [69,75,82,91,112,113]. In six studies participants actively avoided difficult traffic situations and roads that were difficult to walk on, sometimes also because of weather conditions [78,90,93,100,112,113]. Changes in the environment, such as roadworks or unexpected changes of their perception could result in unfamiliarity and confusion [69,91,108]. At these moments people tried to regain familiarity by seeking reference points such as bridges. Road signs and maps can be difficult to understand [69,108]. Furthermore, crowded places and noise levels could create feelings of insecurity [69], and also areas, where there are no people in sight, generated these feelings [97]. One study mentions participants who gradually avoided going outside because they were afraid of falling [74]. Going outside to feel connected with nature is mentioned in five studies [77,83,93,96,113]. For instance, by walking in parks, hearing birds, meeting animals and seeing trees and flowers.

#### 3.3.6. Sphere 6: Experiences with Healthcare and Social Services

Information regarding experiences with healthcare and social services is described in 34 articles. On the one hand a few studies mention the importance of professional support to overcome challenges [94,99], while on the other hand in several studies participants shared that post-diagnostic support did not match their needs [69,95,98,100,101,107,109,115]. Participants wanted to learn more about their condition, experienced obstacles in the system, and sometimes overprotection. Furthermore, participants with young-onset dementia and those living alone missed tailored services [76,100,107].



Experiences with healthcare

Experiences with healthcare are reported in 16 studies. In eight studies participants shared experiences related to receiving information and follow-up after diagnosis. Four studies describe general practitioners and neurologists taking the time to talk with them and their families [84,94,95,101]. In one study participants experienced a lack of information [103]. In five studies participants found that the information was much too biomedical, while negative information induced feelings of incompetence [83,88,94,101,122]. “It took me a few months myself to realise, actually dementia isn’t a death sentence and there is plenty of fun still to be had” [94] (p. 6).

In ten studies participants mentioned their experiences with home care services. In a few studies participants shared positive experiences, and especially participants living alone enjoyed it when caregivers took time to a talk or walk [95,96,97]. These visits could be the only social contact during the day. In five studies participants shared difficulties about the care relationship, for instance, that caregivers were too task-oriented, while they longed for social interaction, took over their autonomy, or that there were too many different caregivers [78,95,106,111,114]. Specific challenges mentioned were struggles with the telecare alarm service that did not function adequately [66] and the desire that homecare workers monitored mealtimes [108]. In two studies participants were not aware of, or could not remember, the reason caregivers visited them, so they suggested that caregivers wrote down when they came and why [84,114].



2.Experiences with social services

Social services, in this section, is used as an umbrella term for services in social care, welfare and social work, individual social work, support groups, community services and respite care. Eighteen studies report experiences with social services.

Individual social work is scarcely mentioned. In only one study a participant talked about a support contact person who offered assistance with everyday tasks, stimulated activities and provided company [117]. This participant felt strengthened, and this contact was even a substitute for old friends. In two studies participants with young-onset dementia reported inadequate support for financial problems after losing their jobs [84,92]. In two studies support groups are mentioned, either tailored to early-stage memory loss [102] or to people with young onset dementia [101]. These programs provided concrete strategies and support for developing a sense of independence and empowerment. In nine studies participants shared their experiences with activity groups of community centres, dementia cafés, day centres and in nursing homes [70,84,86,104,111,112,113,114,120]. Participants mentioned several positive experiences such as feeling enabled to perform activities, having fun in a safe environment, prevention of loneliness and improvement of the spousal relationship by spending time apart. A specific program was developed at a secondary school where participants with dementia attended an adult school, which provided a place to learn new skills and engage in a wider society [121]. In six studies negative experiences regarding group activities are mentioned, such as a lack of day centres close by [83], problems affording the required fee [95], no accommodation for personal interests and capacities [78,88,92] and problems with transportation [88].

#### 3.3.7. Sphere 7: Public Opinions

Information regarding public opinions is described in 11 articles. Participants felt frustrated with the misconceptions of the society about what dementia is and the perceived disabilities [94,107]. They experienced the stigma of being incapable of anything [84,85,95,96,122], as expressed by one of the participants: “When you’ve got Alzheimer’s, everyone thinks that one is just destroyed, which is completely wrong” [96] (p. 16). Negative media portrayals of disempowered people, and the use of obstructive language such as ‘demented’ and ‘sufferer’ were extremely upsetting [94,99,123]. Participants wanted dementia to be normalised in the community. In addition, the media could play a huge role in positively influencing people’s views in that people living with dementia can have a good life [94]. In two studies participants used blogs or Twitter to challenge stigma, raise social awareness, achieve equality and give hope to others with dementia [71,115]. One of the participants expressed: “People don’t realise the positive side, that you can still live, and you can live for quite a long time, depending on the dementia. So, I use it to educate and to change minds about things” [115] (p. 2550). Unfortunately, people were also exposed to being trolled on Twitter, or to receiving tweets questioning their diagnosis [115].

## 4. Discussion

The purpose of this scoping review was to map and describe the evidence on the experiences of individuals living with MCI or dementia in everyday contexts, what challenges them, and supports them. Based on 58 included qualitative studies, seven spheres of everyday life came to the fore: experience of the condition, the self, relationships, activities, environment, health and social services and public opinions. This review provides an extensive look at the individual experiences of disruptions, losses and adjustment to changes in the routine and mundane aspects of daily life of people living with MCI or dementia. Several experiences seem to be specifically linked with the condition of dementia, namely the uncertain future due to the progressive aspect of the disease, the struggle to keep a sense of independence, re-defining identity and experiencing stigma. Numerous strategies and forms of resilience were identified by participants who tried to adapt to these changes. Highlighted in all areas of everyday life is the desire of participants to have reciprocal relationships, to stay engaged through participation in activities and to have a sense of belonging in the community.

This review shows that the change from a clinical care focus to a broader focus on all aspects of everyday life opens rich insights into the insider’s perspective of people living with dementia. This is congruent with the need to better understand how MCI or dementia affects everyday life, and what promotes a meaningful everyday life [21]. The scoping review reveals that there is no consistent definition of everyday life used in the literature on people living with MCI or dementia. Most studies that used the term ‘everyday’ to describe lived experiences did not suggest a definition of everyday life. However, the lens of everyday life creates possibilities to explore these routine and mundane day-to-day experiences from the viewpoint of people living with dementia. To give importance to these experiences also shows that these everyday experiences and practices are more than simply mundane and ordinary [124], as Pink stated: ‘Everyday is at the centre of human existence, the essence of who we are and our location in the world’ [125].

A large number of studies included in this scoping review show the social needs of people living with MCI or dementia. This reflects the social citizenship approach, that people with dementia desire to participate for as long as possible. The results echo the domains of social inclusion, namely the interaction between interpersonal relationships and community participation [126]. In their integrative review, Pinkert et al. [127] suggested that relationships and being integrated into social networks are core aspects of social inclusion for people with dementia. In a Delphi study about what is important for people living with dementia to live well the importance of relationships and meaningful activities were mentioned by 90 per cent of people living with dementia [53]. The findings of this review are in line with the systematic review and meta-analysis of factors associated with quality of life, in the sense that relationships, social engagement and everyday functioning are associated with a better quality of life [128]. The need for reciprocal relationships was also found in a systematic review of the social needs of older people [129]. This seems to reflect a more universal need of being connected to and belonging to a network of social relationships. As such it is likely that being confronted with a chronic and progressive illness such as dementia, makes this even more important. Therefore, it is important to frame dementia as a disability and a human rights concern [130]. However, the other side of the coin of connectedness and social inclusion is loneliness and feelings of exclusion. These experiences are also widely found in the included studies. What is specifically highlighted in this review is the desire for familiarity, which builds upon the literature of ‘at-homeness’: “usually unnoticed, the taken-for-granted situation of being comfortable in, and familiar with, the everyday world in which one lives…” [131] (p. 70). This at-homeness appears to be under pressure when living with dementia.

The different areas of everyday life seem to be closely intertwined and reflect a socioecological model [64,132,133]. As stated, the citizenship approach acknowledges that people are embedded in and shaped by a sociocultural context. Thus, everyday life incorporates both the social and physical environment [134]. This underlines the socio-relational and embodied-material approach described by Ward et al. [135]. Furthermore, the lens of everyday life recognizes humans as social beings, so more attention should be given to the “quality of the social context of everyday life” [136].

Overall, there seems to be much in common between different groups of people living with MCI or dementia when focusing on everyday life experiences. It is noteworthy that many of the experiences are shared by participants of both Western as well as non-Western countries. Nevertheless, in some areas differences between groups have been observed. For instance, people living alone struggled more with feelings of loneliness [77,78,97,100,109,114,117] and people with young-onset dementia missed support for financial problems after losing their jobs [84,92]. It is important to note that the studies did not show a clear difference between people living with MCI or dementia, suggesting that both MCI and dementia have a considerable impact on everyday life. This implies that one of the defining aspects of MCI in the current definition, that symptoms do not interfere with daily life, requires more nuance. In that case it can be argued that the daily experiences of people with MCI need to be taken more seriously in order to support them better, particularly given the distress people feel that their everyday problems could be related to progression into dementia.

### 4.1. Strengths and Limitations

Our review had several strengths. Firstly, the rigorous methodological framework used to explore the existing literature allowed us to present a comprehensive overview of challenges and facilitators of everyday life for people living with MCI or dementia. The findings are widely supported in a large number of studies. Secondly, we focused on the insider’s perspective, i.e., qualitative research representing the authentic voices of people living with dementia [137]. This is important to ensure that support and future research is focussed on what is important to people with dementia and their needs. Thirdly, a large number of qualitative studies were included and reviewed.

This scoping review also had some limitations. Firstly, a wide range of concepts and levels of abstraction were used which made the process of comparing the data difficult. However, in each step of the analysis, the second author was consulted to increase interrater reliability. It is acknowledged that there may be some overlap among categories due to the challenging nature of delineating concepts related to everyday life. Secondly, experiences with healthcare and social services are described, but are likely incomplete as the research question and search strategy were not aimed at mapping the experiences of healthcare and social care. Thirdly, regarding everyday experiences of place, at home and public space, there might be additional relevant search terms that were not identified and used in this scoping review. A fourth limitation is that, although a few articles were found in low- and middle-income countries, most of the included studies were conducted in European countries. This can be due to the selected languages. In the included studies, less attention was given to social determinants such as socioeconomic circumstances and cultural factors. Lastly, this scoping review did not include studies about experiences during the COVID pandemic. The first research results showed that the pandemic has created a greater sense of precarity and tension in how people living with MCI and dementia perceive and experience the outside world [138,139]. It is still unknown what the long-term consequences are, such as the experience of isolation and loneliness.

### 4.2. Implications for Research and Practice

Research on everyday life for people living with dementia adds a new perspective and is an elaboration of everyday citizenship. More research is needed on the factors that promote and impede a sense of reciprocity and belonging for people living with dementia, and how to strengthen reciprocity and belonging within relationships. This requires more qualitative research on the perspective of people living with dementia. Creative and less traditional methods may provide insight and promote inclusive re-search, such as photo elicitation and arts-based methods [137,140,141]. Mobile methods, such as walking interviews, can provide insight in the everyday activities in the social and physical environment [21,104]. These methods are participatory as people are approached as active participants.

The development of a conceptual model for ‘everyday life’ would be beneficial to get more insight in what improves the everyday life experience of people living with MCI or dementia [52]. This would provide a useful foundation for the development of tailored interventions in health and social care as well as to promote an inclusive society. For evaluation of health and social care interventions the Core Outcome Set, developed by Reilly et al. [53], that focuses on what people value in order to live well with dementia, can be useful. Research in facilitators and barriers for the everyday life of people with dementia can be enriched by other perspectives alongside the first-person perspective, namely their next of kin and professionals. Further research ideally could be performed from an intersectional perspective that explores the intersections of various social categories such as culture, social economic circumstances, rural or urban living and sexual orientation [142]. At the same time, in order to implement social innovations, an assessment of the so-called Societal Readiness Level is needed [143].

This review highlights the need for healthcare, social work and policy makers to assess everyday life when working with people living with MCI or dementia. Moreover, a person-centred approach needs to be complemented by a social and community approach. This is in line with the recommendations of a realist review by Li, Keady and Ward [144] who state that that the dynamic relationship between people living with dementia and their everyday neighbourhood have impact on their health and especially social health. An everyday lens implies that in order to foster the resilience of people living with dementia there needs to be an additional focus on neighbourhood and asset-based community interventions alongside individual support [145], in order to promote social inclusion, as stated by the Global Action Against Dementia of the World Health Organization [146].

## 5. Conclusions

Qualitative insights, based on 58 included qualitative studies, in seven relevant categories of everyday life for people living with MCI or dementia were described in this scoping review namely: experiences of the condition, the self, relationships, activities, environment, health and social care and public opinions. Living with cognitive decline entails dealing with progressive disruptions in the mundane and routine aspects of everyday life. This influences the experiences of people with dementia in their relationships, activities, and of the environment. The everyday lens shows a deep desire of people to be connected and stay engaged in a meaningful everyday life. Basic human social needs seem to be under pressure when MCI or dementia enters life, which is also affected by the stigma related to dementia. This highlights the importance of a social citizen approach to care provision and social practice for individuals with MCI and dementia and implies an additional focus on neighbourhood interventions alongside individual support.

## Figures and Tables

**Figure 1 ijerph-19-10828-f001:**
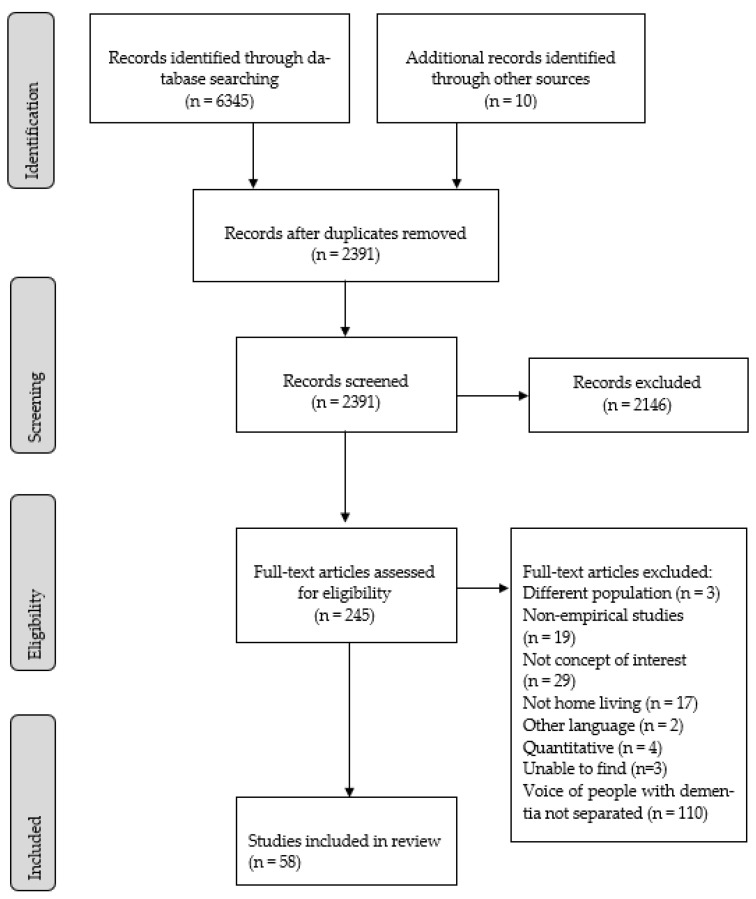
PRISMA flow diagram.

**Figure 2 ijerph-19-10828-f002:**
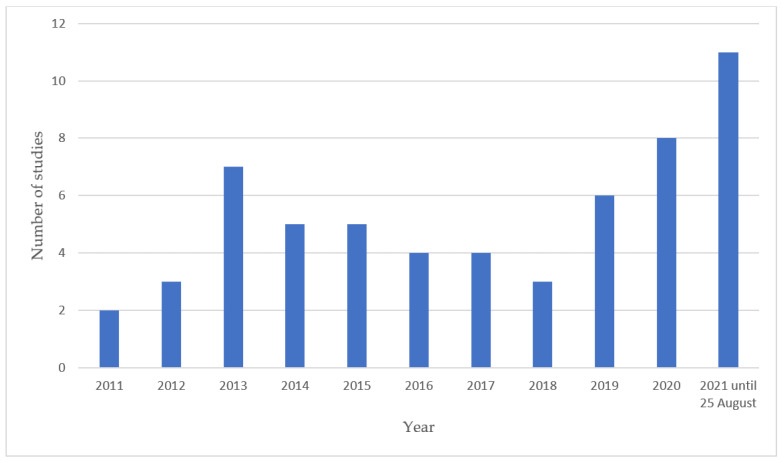
Number of included studies by year of publication.

**Table 1 ijerph-19-10828-t001:** Inclusion and exclusion criteria.

Aspect	Inclusion Criteria	Exclusion Criteria
Participants	People with all forms of dementia (such as Alzheimer’s disease, vascular, frontotemporal and Lewy Body dementia) and pre-dementia, such as mild cognitive impairment (MCI)	The perspectives of next of kin to people with dementia or health and social care professionals.
Concept	Everyday life, synonyms thereof, and related concepts that could give insight into (parts of) daily life.	Measurements using ‘activities of daily living’ questionnaires Intervention research Studies investigating circumscribed events or experiences
Context	Living at home, in the community, or assisted living facilities	Living in long-term care residences such as nursing homes
Type of studies	Qualitative studies (from 2011) that meet the inclusion criteria to capture participants’ subjective experiences and qualitative parts of mixed-method studies Studies published in English, German and Dutch	Quantitative studies Systematic (and other) reviews Conference abstracts and editorials Non-empirical studies and grey literature Studies published in other languages Studies where it is not possible to disentangle the perspective of people living with dementia

**Table 2 ijerph-19-10828-t002:** Search terms of studies.

Set	Category	Search Term
S1 AND	Participant	dementia OR Alzheimer Disease OR Lewy body OR vascular dement * OR frontotemporal dement * OR mild cognitive impairment OR MCI
S2 AND	Concept	daily life OR daily living OR day to day OR everyday OR living well OR lifeworld OR lived experience OR life experience
S3	Study type	qualitative OR phenomenol * OR ethnograph * OR grounded theory OR experience * OR interview OR photo

Note: The truncation symbol (*) is used as a substitute for any string of zero or more characters in the search term.

**Table 3 ijerph-19-10828-t003:** Summary of study characteristics.

	Number of Articles
Country:	
Asian countries	2
Australia	2
Belgium	2
Middle East	1
Scandinavia	21
South America	2
Switzerland	1
The Netherlands	4
United Kingdom (UK)	18
United States of America (USA)/Canada	5
Not reported	1
Age:	
People with young onset dementia	12
Older people (>65 years)	23
Mixed	23
Stage:	
Mild cognitive impairment	6
MCI and dementia	2
Dementia	50
Concept:	
Everyday life	0
Lived experiences, lifeworld	10
Part of everyday life or another concept	48
Data collection methods:	
Face-to-face interviews	51
Telephone interviews	1
Focus groups	5
Blogs	1

**Table 4 ijerph-19-10828-t004:** Description of the seven spheres of everyday life.

Sphere	Description
1. Dementia	How people experience consequences related to the condition in their life
2. Self	How people evaluate themselves personally and how they experience their emotions and body
3. Relationships	How people experience relationships
4. Activities	How people experience everyday activities
5. Environment	How people experience their environment
6. Healthcare and social services	How people describe experiences with healthcare and social services
7. Public opinions	How people experience public opinions with regard to dementia

**Table 5 ijerph-19-10828-t005:** Extraction of categories and subcategories.

References	1. Dementia	1.1. Receiving Diagnosis	1.2. Changes	1.3. Future	2. Self	2.1. Self-Evaluations	2.2. Emotions	2.3. Body	3. Relationships	3.1. Partner/Family	3.2. Friendships	3.3. Neighbourhood	4. Activities	4.1. Activities of Daily Living	4.2. Participation	5. Environment	5.1. Home	5.2. Public Environment	6. Health and Social Care	6.1 Healthcare	6.2 Social Care	7. Public Opinions
Bartlett (2019) [66]			x		x	x			x		x	x	x		x							
Berg (2013) [67]	x		x	x					x	x	x		x		x				x	x		
Borley (2016) [68]	x	x			x	x			x	x			x	x	x							
Brorsson (2011) [69]													x	x	x	x		x	x			
Buggins (2021) [70]	x	x	x	x					x		x		x		x				x		x	
Castaño (2019) [71]	x	x	x		x	x	x		x	x	x											x
Cedervall (2015) [72]	x			x	x	x	x	x					x		x							
Chen (2019) [73]	x		x		x	x		x	x	x												
Chung (2019) [74]	x	x			x	x		x					x	x	x			x				
Clark (2020) [75]					x	x	x		x	x	x	x	x	x		x		x				
Clemerson (2014) [76]	x	x		x	x	x	x	x	x	x	x								x			
Dooley (2021) [77]	x		x	x	x		x		x	x			x	x	x	x		x				
Duane (2011) [78]	x	x			x	x		x	x	x			x	x	x	x	x	x	x	x	x	
Frazer (2011) [79]	x			x	x	x	x	x	x		x	x	x	x								
Griffin (2016) [80]	x	x			x	x		x	x	x	x		x	x	x							
Hedman (2016) [81]													x	x	x	x	x					
Hellström (2015) [82]													x	x		x	x	x				
Hicks (2021) [83]	x		x		x	x		x	x	x		x	x	x	x	x	x		x	x	x	
Johannessen (2013) [84]	x	x			x	x			x	x			x	x	x				x	x	x	x
Johannessen (2014) [85]	x			x	x			x	x		x											x
Johannessen (2019) [86]	x	x	x	x	x			x					x	x	x				x		x	
Johansson (2011) [87]													x	x								
Johansson (2015) [88]	x		x		x			x	x	x	x	x	x	x	x				x	x	x	
Lin (2021) [89]	x		x		x	x	x	x	x	x	x		x		x							
Lloyd (2015) [90]													x	x	x	x	x	x				
Margot-Cattin (2021) [91]													x		x	x		x				
Mazaheri (2013) [92]	x		x		x	x			x	x	x		x		x				x		x	
McDuff (2015) [93]					x			x					x	x	x	x		x				
Mitchell (2020) [94]					x	x			x	x			x	x	x				x	x		x
Moe (2021) [95]	x	x	x		x	x		x	x		x		x	x	x				x	x	x	x
Odzakovic (2020) [96]	x			x	x	x		x	x	x	x	x	x	x	x	x	x	x	x	x		x
Odzakovic (2021) [97]	x	x							x	x	x	x	x		x	x		x	x	x		
Parikh (2016) [98]	x		x	x	x	x	x		x	x	x		x		x							
Pipon-Young (2011) [99]	x	x		x					x	x	x	x	x	x	x				x			x
Portacolone (2018) [100]	x	x	x	x	x		x	x	x	x	x		x	x		x		x	x			
Rabanal (2018) [101]	x	x	x		x		x		x	x	x		x	x	x				x	x	x	
Renn (2021) [102]	x	x	x	x	x			x	x	x	x	x	x	x	x				x		x	
Roberts (2013) [103]	x	x	x		x	x	x		x	x	x								x	x		
Robertson (2014) [104]					x	x			x	x	x		x		x				x		x	
Robinson (2012) [105]	x	x	x	x	x	x	x		x	x												
Rostad (2013) [106]	x	x			x	x			x		x		x	x		x	x		x	x		
Sakamoto (2017) [107]	x		x		x	x													x			x
Sandberg (2017) [108]									x			x	x	x		x		x	x	x		
Steeman (2013) [109]					x	x			x				x	x					x			
Steenwinkel (2014) [110]	x		x	x	x	x	x		x				x	x	x	x	x		x	x		
Strandenæs (2017) [111]																			x		x	
Sturge (2020) [112]									x	x	x		x	x		x		x	x		x	
Sturge (2021) [113]									x	x	x	x	x	x	x	x		x	x		x	
Svanström (2015) [114]	x		x		x	x		x	x		x		x	x	x	x	x		x	x	x	
Talbot (2021) [115]					x	x			x		x		x		x				x			x
Thoft (2020) [116]	x	x	x						x	x	x		x	x	x							
Thorsen (2020) [117]	x		x		x	x	x	x	x	x	x	x	x	x		x	x		x		x	
Trindade (2018) [118]					x		x		x	x			x	x								
Trindade (2020) [119]	x		x		x		x	x	x	x			x	x	x							
Vliet (2017) [120]	x	x			x	x			x	x	x	x	x		x				x		x	
Ward (2020) [121]									x	x	x		x	x					x		x	
Wijngaarden (2019) [122]	x	x	x	x	x	x		x	x	x	x		x	x	x	x			x	x		x
Xanthopoulou (2019) [123]	x	x	x	x	x	x	x		x	x	x	x	x	x	x							x
Frequency (%) (*N* = 58)	39	22	26	17	32	32	18	21	46	35	33	14	50	38	38	21	10	15	34	16	18	11

## Data Availability

Not applicable.

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
