# Peer review of "Everyday Experiences of People Living with Mild Cognitive Impairment or Dementia: A Scoping Review"

_ijerph, 2022, doi:10.3390/ijerph191710828_

Round 1
Reviewer 1 Report
The submitted article is about review of dementia and everyday experiences of people living methods with cognitive impairment. The article is well structured and the results are well presented and discussed. However, following comments will be helpful for further improvement:
1. There some grammatical error in the manuscript which need to be addressed.
2. Authors can also discuss the further research direction to improve the everyday life experience of people living with dementia.
3. Authors can also provide a graphical presentation of selected research articles along with year of publication in Selection of Evidence such as given in https://doi.org/10.1155/2022/9288452
Reviewer 2 Report
Thank you for allowing me to review this thoughtful, rigorous and well-written manuscript.
This manuscript presents a scoping review of the everyday experiences of people living with dementia. I affirm that this is an important topic that needs to be addressed in order to promote positive outcomes for this population, and appreciate the efforts of the authors. I believe it is well-suited for the journal, specifically the current special issue. Below are some comments and suggestions that arose when reading the article that I believe will help strengthen it. I hope you find them useful!
1. Lines 40-42: The authors comment on the deficit model of dementia discourse. Is there any research promoting the opposite, i.e., treatments that focus on their strengths and what they still can do? Why do you think that Western society historically lands on the deficit model? If so, it is worth mentioning here for contrast.
2. Lines 53-77: The authors comment on 2 important theories/social concepts. Have they been studied in other health conditions? How would this paradigm shift change health outcomes, dementia or otherwise? Data with evidence supporting these models compared the unfortunate traditional methods such as the deficit models would be interesting to have here, as it backs up the points the authors are trying to make through the rest of the manuscript.
3. Table 1: Inclusion criteria includes all forms of dementia (AD, Lewy body, etc.). Did the authors notice any differences based on dementia type? I say this as Lewy Body dementia is a common consequence in Parkinson’s disease, which is known to have its own independent challenges (many of which fall into the domains of interest in this paper) that may exacerbate the experience.
4. I appreciate the thoroughness and thoughtfulness of the methods section. Well done! While it may not fit within the scope of this manuscript, it would be interesting to follow up with differences in personal experiences based on country (since you have various countries reported in Table 3), as cultural norms tend to influence care.
5. The results of the scoping review were well-written and organized in a way that flowed well.
6. The discussion summarizes findings well. This would be another great place to cite research that does implement this type of person-centered care, cognitive-related or otherwise to support the notion that it promotes positive health outcomes, whether it be onset/progression of the disease or increased health-span (see point #2). Toward the end, or in the “implications for research and practice” section, it would be interesting for the authors to hypothesize what outcomes would look like for this population with patient-centered care as discussed in the manuscript.
7. The authors mention that “more research is needed” throughout the document. What type(s) of research would the authors warrant to be appropriate based on their knowledge and what is presented in the manuscript?
8. You may also argue that the degree to which these domains are affecting PLWDs lives may also be greater than we realize, given that the ability to communicate/articulate diminishes as the disease state progresses, so there may not be adequate reporting on severity.
